# Impact of mindfulness on football coaches: A mixed design

**Aleix Gibert**[1]*, **Antoni Planas**[2], **Carlota Torrents**[1,2]

**1** Complex Systems and Sport Research Group (Spain), Barcelona, Spain, **2** Institut Nacional d'Educació Física de Catalunya (INEFC), Universitat de Lleida, Lleida, Spain

* aleix.gibert.buxeda@gmail.com

**Data Availability Statement:** All relevant data are within the manuscript and its Supporting information files.

## Abstract

Mindfulness is an increasingly popular practice among elite athletes to improve performance and well-being, but its application for coaches is still very limited. Therefore, a new program (M×E; Mindfulness para Entrenadores) was designed and implemented online for 6 weeks for coaches, support staff, and technical directors. Participants were 58 (57 men), aged between 23 and 58 (*M* = 31.8). The between-subject analysis (RM ANOVA 2×2) included experimental (*n* = 26) and wait-list control group (*n* = 21). A total of 29 completed the within-subject analysis (RM ANOVA) completing follow-up measurements until six months. The analysis was complemented with 21 individual semi-structured interviews. The experimental group showed significant improvements in mindfulness trait (*p* < .001), interpersonal mindfulness (*p* = .010), and a significant improvement in emotional regulation (*p* = .010) in comparison to the wait-list control group. The experimental group's positive trend in all variables' levels was maintained until six months after the program. The qualitative analysis showed a positive impact on professional, personal, and social areas with improvements related to performance and well-being. Several participants considered the program as "fundamental" for their professional role. This study provides information on best practices in implementing mindfulness-based programs, highlighting their practical orientation, the training plan, and the safe space. These results offer initial validation of the potential of the M×E and invite sports organizations to incorporate mindfulness-based programs specially designed for coaches, support staff and technical directors.

## Introduction

Although coaches appear to be an ideal population for interventions, considering their essential role and influence on the athlete [1], the target of the research focused on training interventions has historically been the athlete. For instance, while mindfulness is an increasingly popular practice among elite athletes to improve performance and well-being [2], their effect on coaches remains largely unexplored. Although few, mindfulness interventions for coaches showed encouraging results such as stress reduction, improved well-being, improved interactions with athletes [3], better recovery, and burnout prevention [4].

Coaches ability to manage stress is critical as it is recognized that pressure to succeed, excessive workload, lack of job security, frequent travel, and isolation [5] can negatively affect their

**Funding:** This work was supported by the National Institute of Physical Education of Catalonia (INEFC).

**Competing interests:** The authors have declared that no competing interests exist.

performance and long-term health [6]. However, coaches are generally not well equipped to handle stress, high levels of anxiety, fatigue, tension, anger, and depressive emotions [7]. The ability to deal with emotions is a fundamental skill for coaches as it is related to improved performance, player satisfaction, good relationships, and effective leadership [8].

Correlational research supports the association between mindfulness and reduced difficulties in emotion regulation [9]. Mindfulness practice can improve emotion regulation skills by decreasing both over-engagement (e.g., rumination) and under-engagement (e.g., avoidance) and facilitating healthy, adaptive engagement that promotes clarity and functional use of emotional responses [10].

Mindfulness training also allows coaches to be more present and improve concentration [11]. Specifically, mindfulness training has been positively related to selective attention [12], sustained attention [13], the orientation of attention [14], and attentional flexibility [15]. Similarly, Kee (2019) proposed that a coach trained in mindfulness would be in a better position to appreciate the nuances of emergent behavior [16]. In particular, by adopting an open awareness of the player's behavior and temporarily suspending any judgment, and pointed out the relevance of interpersonal mindfulness.

Interpersonal mindfulness describes the capacity of people to be mindful during their interpersonal interactions [17]. Literature about the importance of interpersonal skills for coaches is vast. For instance, Côté and Gilbert (2009), in their integrative definition of coaching effectiveness, highlighted the importance of a coach's ability to create and maintain relationships and encouraged coaches to look for methods through which they can continually develop their interpersonal knowledge in order to communicate appropriately and effectively with their athletes [18]. Importantly, positive communication styles have improved athletes' mental health [19], and psychologically safe environments [20].

Finally, it's considered that the coach's ability to maximize athletes' results is also based on constant introspection [18]. Giges et al. (2004) postulated that self-aware coaches could more effectively assess and respond to their players, better see what effect their behaviors, feelings, or communication have on their players, and understand what generates their thoughts, feelings and actions [21]. Even though the importance of self-regulation and interpersonal and intrapersonal knowledge for coaches, the education programs have generally focused just on the professional knowledge [22].

For all these reasons, this study aimed to assess the effects of a new mindfulness-based program intended for coaches, support staff, and technical directors on the levels of mindfulness trait, interpersonal mindfulness and difficulties in emotional regulation. The hypothesis was that the participants would increase their levels of mindfulness trait and interpersonal mindfulness and decrease their difficulties in emotional regulation. Furthermore, this study aimed to bolster best practices for implementing mindfulness-based programs for this population.

## Method

The research project was carried out in an elite football academy, headquartered in Barcelona (Spain), and includes several international academies. The program was offered to coaches, support staff, coordinators and leadership positions such as technical directors. The study was developed following the guidelines of the Declaration of Helsinki and the Guide to Good Research Practice. The protocol was approved by the Clinical Research Ethics Committee of the Sports Administration of Catalonia (21/CEICGC/2020). Study information was provided orally and in writing to the participants, who could ask questions before agreeing to participate. Confidentiality and anonymity were maintained in accordance with Law 15/1999. The

data was coded and only the research team had access to it. The recruitment process started on February 5$^{th}$ of 2021 and finished on May 20$^{th}$ of the same year.

## Design

To analyze the impact of the mindfulness-based program on the levels of mindfulness trait, interpersonal mindfulness, and difficulties in emotional regulation, a quasi-experimental and longitudinal design was used. The experimental and wait-list control groups completed the pre- and post-tests, but only the experimental group completed the follow-up measurements.

## Participants

A total of 58 participants took part in the study. The inclusion criterion was to be part of the Academy. The age of the participants ranged from 23 to 58 years ($M = 31.8$; $SD = 6.2$), among which there was only one woman, a proportion in terms of gender that coincides with the internal reality of the Academy. The vast majority were Caucasian (97.9%); 41.4% of the participants had "no" previous experience with mindfulness or meditation, 48.3% had meditated "sometimes", and the remaining 10.3% had previous experience. Participants included coaches (39.7%), technical directors (34.5%), coordinators (12.1%), and members of the Methodology Department (8.6%), and completed the sample two project managers, one physical trainer, one tactical analyst, and one team delegate. Regardless of the current role, except for two participants, all had several years of experience as a football coach ($M = 10.2$; $SD = 5.1$).

Participants who agreed to participate in the program were separated into an experimental or wait-list control group: 47 completed the pre- and post-test (experimental group, $n = 26$; wait-list control group, $n = 21$). Right after, the program was carried out with the wait-list control group. Finally, 29 participants completed the follow-up measurements.

## Procedure

The club approved the proposal with a complete description of the study. Afterwards, all Academy members were summoned to a 30-minute online presentation in which the study's purpose, the program's characteristics, and the voluntary nature of their participation were presented. Subsequently, an online form was sent so that those interested could register. The homogeneous distribution in different time slots allowed the distribution of the participants between an experimental and wait-list control group. The study information and informed consent were delivered, duly filled out and signed. The evaluation instruments were administered online with a clear and equal procedure for all participants. The program was carried out online. In the week following the end of the program, the levels of the different variables in both groups were re-evaluated. Subsequently, the wait-list control group did the program as agreed. Finally, two follow-up measurements (at 3 and 6 months) were administrated to those participants who had completed the program.

A new mindfulness-based program was designed (M×E; Mindfulness para Entrenadores), combing those contents considered most relevant from well-established programs such as Mindful Sports Performance Enhancement (MSPE) [23], Mindfulness Acceptance and Commitment (MAC) [24], and Mindfulness-based Stress Reduction (MBSR). The program consisted of 6 weeks with a weekly session of 75 min. at the express will of the club. The first three sessions were adapted from the MSPE: S1 presented the fundamentals of mindfulness, S2 focused on the states of maximum performance and their facilitators, and S3 encompassed those contents considered most relevant from the other MSPE sessions. S4, inspired by the MBSR, had emotions as its central theme, including self-regulation strategies; S5 presented the role that values can play as a compass for behavior and was adapted from the work of Josefsson

**Table 1. M×E's contents.**

| Session | | | Training plan | | | |
|---|---|---|---|---|---|---|
| Title | Contents | In-class practice | Formal practice | Informal practice | Professional practice | Self-awareness practice |
| S1<br>The fundamentals of mindfulness | Introduction, confidentiality and safe space<br>Program's contextualization and justification<br>Definition of mindfulness<br>The "autopilot"<br>Attention training<br>Explaining the program's dynamic | M1<br>(10 min.) | 5x M1 | Moments STOP!<br>+<br>Eat consciously | | Autopilot |
| S2<br>States of maximum performance | Introduction: practice as landing<br>Dynamic: Define your peak performance states<br>The relationship between mindfulness and flow states<br>Performance facilitators and strategies for the coach<br>The importance of practice and tips to create the habit | M1<br>(10 min.) | 5x M1<br>+ 1x BS | Moments STOP!<br>+<br>Shower consciously | Field: Warm-up<br>Office: 2 blocks 45/15<br>All: propose anchors | Peak performance states |
| S3<br>Principles and concepts | Recognizing the power of expectations<br>Acceptance of "what it is"<br>Acceptance vs. resignation / process vs. result<br>Limiting beliefs | M2<br>(10 min.) | "1 per day"<br>5x M2 or M1<br>1x BS<br>+ 1x CM | Moments STOP!<br>+<br>Brush your teeth consciously | Field: Warm-up + 1st drill<br>Office: 2 blocks 50/10<br>All: check anchors | Limiting beliefs |
| S4<br>The emotions | Dynamic: Basic emotions: fear, anger, sadness and joy<br>Adaptive and non-adaptive emotions<br>Emotional regulation strategies<br>Stress: prevention and regulation strategies | M2<br>(10 min.) | "1 per day"<br>5x M<br>1x BS<br>1x CM | Moments STOP!<br>+<br>Walk consciously | Field: Warm-up + 1st and 2nd drill, and 1/2 Match<br>Office: 2 blocks 60/15<br>All: check anchors | Emotions |
| S5<br>Values, objectives, behaviors and commitment | Introduction of the MAC program<br>Dynamic: the arrow<br>Values-based behavior vs. based on emotions<br>The club's values: looking for individual and collective alignment<br>Commitment | M3<br>(10 min.) | "1 per day"<br>4M<br>1x M3<br>1x BS<br>1x CM | Moments STOP!<br>+<br>Drive consciously | Field: Whole training<br>Office: 2 blocks 75/15<br>All: check anchors | Values and behavior |
| S6<br>Conscious communication + the end of the beginning | Dynamic: active listening<br>Communication styles: aggressive, passive, assertive<br>Guidelines for assertive communication<br>The end of the beginning: summary and guidelines for continuity | M3<br>(10 min.) | Create your own plan | | Field: Whole Training+Match<br>Office: 2 blocks 90/20<br>All: define your anchors | Communication |

Note. All sessions (except S1) began with guided meditation and continued with discussion of home practice. Except for the last one, all of them ended with the explanation of the weekly training plan. M = meditation of minimum 5 min.; M1 = diaphragmatic breathing + sitting meditation: attention to breathing (10 min.); M2 = sitting meditation: attention to breathing and the body as a whole (15 min.); M3 = sitting meditation: attention to breathing, body and environment (20 min.); BS = body scan (17 min.); CM = conscious movement.

et al. (2019) using the MAC [25]; to end with, S6 included the theme of conscious communication (also inspired by the MBSR), and ended with a summary of the main contents and recommendations to continue with the practice. The primary characteristics of mindfulness (attention in the present moment and the attitude of acceptance) were present cross-sectionally. The reader can consult each session's contents and formal practices in Table 1.

The M×E provided a weekly training schedule that progressively increased both in volume and variability, with four sections: (i) formal practice, (ii) informal practice, (iii) professional practice, and (iv) self-awareness practice. Formal practice included three sitting meditations lasting 10, 15, and 20 minutes, a body scan, and mindful movement. Informal practice focused on practicing mindfulness in daily life as well as STOP moments (Stop, Breathe, Observe, Proceed). Professional practice, designed specifically for this program, aimed to transfer mindfulness to work-related scenarios. For instance, the on-pitch training plan started with being mindful during the warm-up and progressively increased volume and variability, ending with practicing being mindful throughout the whole training session. Finally, self-awareness practice proposed a weekly objective related to the content (for example, be aware of the autopilot in week 1).

Microsoft Teams was used as an online platform (the one used by the club). The first author, who had extensive experience with mindfulness and meditation as well as the leadership on mindfulness-based programs, was the instructor, being a good connoisseur not only of the content but also of the context of the participants, with previous experience in most participants' roles (coach, coordinator, and technical director) in the same club. The club's sports psychology team supervised the program.

## Instruments

A mixed design with quantitative and qualitative data collection was used to analyze the effects of the mindfulness-based program. The levels of mindfulness trait, interpersonal mindfulness, and difficulties in emotion regulation were evaluated through questionnaires. At the end of the program, individual semi-structured interviews were conducted. The instruments used were the following:

**Background Questionnaire.** This instrument was administered only before the intervention. Participants reported their age, gender, role in the club, years of coaching experience, and experience with mindfulness and meditation practice.

**Spanish Version of the Five Facet Mindfulness Questionnaire (FFMQ).** The FFMQ is a 39-items rated on a 5-point Likert-type scale ranging from 1 (*never* or *very rarely true*) to 5 (*very often* or *always true*) [9]. It includes five subscales: observing, describing, acting with awareness, non-judgment of one's own experience, and nonreactivity to inner experience. The short Spanish version used (FFMQ-SF) [26], with 20 items, presents good psychometric reliability and validity in its five subscales, being an appropriate questionnaire for evaluating mindfulness in a non-clinical population.

**Interpersonal Mindfulness Scale (IMS).** The IMS [27] is a 27-item scale rated on a 5-point Likert scale ranging from 1 (*almost never*) to 5 (*almost always*). It includes four subscales: (i) presence, (ii) awareness of self and others, (iii) acceptance without prejudice, and (iv) non-reactivity. It was developed to assess mindfulness during interpersonal interactions. It presents strong psychometric, reliability, and internal validity properties [28].

**Spanish Version of The Difficulties in Emotion Regulation Scale (DERS).** The DERS [29] is 36-items scale rated on a 5-point Likert scale ranging from 1 (*almost never*) to 5 (*almost always*). Higher scores indicate greater difficulties in regulating emotions. It was designed to assess six specific subscales of emotional dysregulation: (i) awareness, (ii) clarity, (iii) drive, (iv) goals, (v) non-acceptance, and (vi) strategies. The Spanish version [30] presents an internal consistency of the subscales from moderate to satisfactory, except for awareness, which was low.

**Practice-log.** This instrument was used by the participants to keep a daily record of their practice, noting the exercise and duration of each practice and adding any relevant observation such as difficulties or elements from the experience itself.

**Final interview.** A semi-structured interview was designed to deepen the participants' subjective assessment in two main dimensions: (i) the impact of the program, and (ii) elements of the program, with the aim of continuing to improve the program's design. According to Sparkes and Smith (2015) [31], the interview is the best instrument to obtain detailed and complete qualitative information. The questions were previously validated through an expert's review both in the subject and in qualitative research methodology. A total of 21 participants (43.8% of the total) completed it and provided a representative sample of the participants. All interviews were carried out online with the presence of the interviewee and the first author, had an average duration of 21 minutes and were recorded with the consent of the participants for later transcription.

## Analysis

First, the questionnaires were analyzed for missing or out-of-range values. Subsequently, descriptive analyzes were performed to summarize the characteristics of the study population. The IBM SPSS Statistics 28.0 was used to carry out all statistical analyses, and statistical significance was set at $p < .05$.

**Between-subject analysis.** The normality in the distribution of the variables was confirmed using the Shapiro-Wilk test. Therefore, to summarize the quantitative variable, the mean ($M$) and its respective standard deviation ($SD$) are presented at each moment the measurement was carried out. The homogeneity of variances or sphericity was confirmed using the Mauchly test, so parametric tests were applied. First, pre-test levels of both groups were analyzed through $t$ test for independent samples to confirm that there were no significant differences prior to the program. Next, 2×2 repeated-measures (RM) ANOVA was applied, where experimental "group" was used as a between-subjects factor and "time" as a within-subjects factor (i.e., pre-test and post-test). In case of detecting significant effect interactions (group × time) in any of the variables, follow-up RM ANOVAs for each subscale were performed. Finally, exhaustive comparisons (Post Hoc) were performed for the global scores, applying the adjustments for multiple comparisons (Bonferroni).

**Within-subject analysis.** In addition, the evolution of the impact of the program in the experimental group was analyzed up to 6 months after the intervention. The normality in the distribution of the variables was confirmed using the Shapiro-Wilk test. Mauchly's test confirmed sphericity, so parametric tests were applied. The study hypotheses were evaluated through RM ANOVA using "time" as a within-subjects factor. In case of detecting significant effects in any of the variables, follow-up RM ANOVAs for each subscale were performed. Finally, exhaustive comparisons (Post Hoc) were performed for the global scores, applying the adjustments for multiple comparisons (Bonferroni).

**Qualitative analysis.** It was done using thematic analysis, a method for identifying, analyzing, and reporting patterns (themes) within data [32]. Six steps were followed: (i) familiarization with the data; (ii) generation of the initial codes based on the purpose of the study; (iii) grouping the codes into main themes; (iv) examination of the topics; (v) definition and names of topics; and (vi) preparation of the report. Recording of the interviews and verbatim transcription ensured descriptive validity. Two collaborators with experience in the subject reviewed the categorization of each code.

## Results

### Feasibility and acceptability

The 76.4% of participants who completed the program attended all six sessions, 20% attended at five, and the remaining 3.6% attended at four. Nine participants did not complete the

**Table 2. Means, standard deviations, and repeated-measures ANOVA interaction effect in all dependent variables and subscales.**

| Variable | Experimental | | | | Wait-list control | | | | | | |
|---|---|---|---|---|---|---|---|---|---|---|---|
| | Pre-test | | Post-test | | Pre-test | | Post-test | | $F_{(1, 45)}$ | $p$ | $\eta_p^2$ |
| | M | SD | M | SD | M | SD | M | SD | | | |
| Mindfulness trait | 3.399 | 0.509 | 3.642 | 0.415 | 3.503 | 0.449 | 3.433 | 0.464 | 15.082 | .000*** | .251 |
| Observing | 2.992 | 1.032 | 3.350 | 0.828 | 3.238 | 0.923 | 3.362 | 0.862 | 1.315 | .258 | .028 |
| Describing | 3.377 | 0.801 | 3.496 | 0.706 | 3.529 | 0.654 | 3.519 | 0.613 | 0.719 | .401 | .016 |
| Acting aware | 3.319 | 0.749 | 3.542 | 0.723 | 3.438 | 0.778 | 3.324 | 0.926 | 4.443 | .041* | .090 |
| Non-judging | 3.885 | 0.686 | 4.092 | 0.645 | 4.005 | 0.769 | 3.710 | 0.677 | 8.834 | .005** | .164 |
| Non-reaction | 3.423 | 0.671 | 3.727 | 0.574 | 3.305 | 0.590 | 3.252 | 0.581 | 6.130 | .017* | .120 |
| Interpersonal mindfulness | 3.689 | 0.437 | 3.829 | 0.489 | 3.715 | 0.318 | 3.575 | 0.340 | 7.319 | .010* | .140 |
| Presence | 3.523 | 0.438 | 3.565 | 0.428 | 3.481 | 0.445 | 3.386 | 0.498 | 1.100 | .300 | .024 |
| Awareness of self and others | 3.788 | 0.540 | 4.035 | 0.578 | 3.929 | 0.434 | 3.790 | 0.492 | 9.753 | .003** | .178 |
| Nonjudgmental acceptance | 3.715 | 0.660 | 3.854 | 0.693 | 3.819 | 0.551 | 3.576 | 0.602 | 4.938 | .031* | .099 |
| Nonreactivity | 3.731 | 0.575 | 3.862 | 0.543 | 3.633 | 0.542 | 3.548 | 0.578 | 1.917 | .173 | .041 |
| Difficulties in emotional regulation | 2.205 | 0.544 | 2.058 | 0.543 | 2.348 | 0.593 | 2.418 | 0.549 | 6.478 | .014* | .126 |
| Awareness | 2.658 | 0.734 | 2.385 | 0.536 | 2.757 | 0.506 | 2.686 | 0.523 | 1.778 | .189 | .038 |
| Impulse | 1.946 | 0.730 | 1.792 | 0.655 | 2.033 | 0.653 | 2.157 | 0.618 | 7.351 | .009** | .140 |
| Non-acceptance | 2.131 | 0.619 | 1.796 | 0.734 | 2.319 | 0.910 | 2.248 | 0.771 | 1.796 | .187 | .038 |
| Goals | 2.800 | 0.835 | 2.646 | 0.920 | 2.952 | 1.022 | 3.000 | 0.925 | 1.047 | .312 | .023 |
| Clarity | 1.969 | 0.582 | 1.923 | 0.546 | 1.914 | 0.622 | 2.219 | 0.713 | 6.241 | .016* | .122 |
| Strategies | 1.727 | 0.706 | 1.808 | 0.729 | 2.114 | 0.727 | 2.200 | 0.744 | 0.002 | .966 | .000 |

*Note.* For these analyses, experimental condition $n = 26$, and wait-list control condition $n = 21$. Main effects do not relate to the study hypothesis and are, therefore, not presented. Interested readers may contact the authors for details.

$\eta_p^2 = .01$ = low effect size; $\eta_p^2 = .06$ = medium effect size; $\eta_p^2 = .14$ = large effect size.

*$p < .05$.

**$p < .01$.

***$p < .001$.

minimum of four sessions and therefore were excluded. Participants reported weekly practice throughout the program for 5.13 days a week on average ($SD = 0.88$, range = 2–7). Most participants (92.6%) responded that they would like to continue the training.

## Comparison between experimental and control groups

The descriptive statistics by experimental condition at pre- and post-test are presented in Table 2. Independent samples *t*-tests showed no significant baseline differences between groups. RM ANOVA 2×2 revealed a statistically significant interaction between the effects of condition and time on the global scores of the three variables (mindfulness trait, interpersonal mindfulness and difficulties in emotion regulation). The analysis of the evolution of each group separately is illustrated in Fig 1. It showed a statistically significant difference in the experimental group ($t_{(1, 45)} = 2.579$; $p = .013$; 95% CI 1.843, 2.429), but the difference was not significant in the wait-list control group ($t_{(1, 45)} = 1.111$; $p = .276$; 95% CI 2.100, 2.658). Consequently, additional RM ANOVAs 2×2 were performed for each subscale separately.

## Follow-up of the experimental group

RM ANOVAs were used to determine whether the levels of the study variables changed from the pre- to the 6-month follow-up test for the experimental groups. As shown in Table 3 and

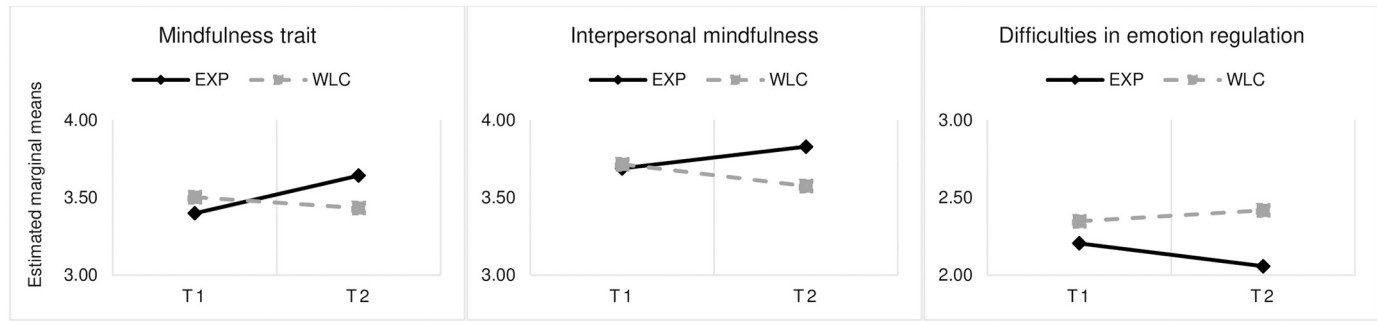

**Fig 1. Estimated marginal means of study variables for each experimental condition at pre and post periods.** Note. For these analyses, experimental condition *n* = 26, and wait-list control condition *n* = 21. T1 = pre-test; T2 = post-test.

illustrated in Fig 2, the analysis revealed that time did have a significant effect, between at least two times, on the global scores of the three variables. Follow-up RM ANOVA for each subscale showed that time had a significant effect on most of them, except for describing (an aspect of mindfulness trait), strategies (an aspect of difficulties in emotion regulation), and presence (an aspect of interpersonal mindfulness) without any significant effect.

In the Post Hoc comparison between moments, the Bonferroni correction was applied. Statistically significant differences were detected in mindfulness trait between T1 (pre-test) and

**Table 3. Means, standard deviations, and repeated-measures ANOVA in all dependent variables for the experimental group.**

| Variable | Pre-test | | Post-test | | 3-month | | 6-month | | $F_{(3, 84)}$ | *p* | $\eta_p^2$ |
|---|---|---|---|---|---|---|---|---|---|---|---|
| | *M* | *SD* | *M* | *SD* | *M* | *SD* | *M* | *SD* | | | |
| Mindfulness trait | 3.463 | 0.517 | 3.748 | 0.420 | 3.848 | 0.420 | 3.827 | 0.417 | 13.875 | .000*** | .331 |
| Observing | 3.043 | 0.918 | 3.436 | 0.789 | 3.369 | 0.766 | 3.476 | 0.838 | 3.337 | .022* | .075 |
| Describing | 3.471 | 0.810 | 3.650 | 0.703 | 3.762 | 0.690 | 3.838 | 0.626 | 2.490 | .063 | .057 |
| Acting aware | 3.333 | 0.847 | 3.569 | 0.705 | 3.983 | 0.586 | 3.874 | 0.627 | 11.706 | .000*** | .222 |
| Non-judging | 3.781 | 0.780 | 4.026 | 0.733 | 4.269 | 0.701 | 4.198 | 0.671 | 5.866 | .000*** | .125 |
| Non-reaction | 3.331 | 0.645 | 3.614 | 0.597 | 3.736 | 0.595 | 3.740 | 0.550 | 6.603 | .000*** | .139 |
| Interpersonal mindfulness | 3.725 | 0.430 | 3.917 | 0.436 | 3.972 | 0.384 | 3.991 | 0.399 | 7.214 | .000*** | .217 |
| Presence | 3.537 | 0.499 | 3.656 | 0.491 | 3.637 | 0.398 | 3.689 | 0.448 | 0.866 | .463 | .032 |
| Awareness of self and others | 3.859 | 0.567 | 4.119 | 0.542 | 4.115 | 0.508 | 4.148 | 0.537 | 8.894 | .000*** | .255 |
| Nonjudgmental acceptance | 3.677 | 0.650 | 3.889 | 0.661 | 4.004 | 0.621 | 4.048 | 0.605 | 6.006 | .000*** | .188 |
| Nonreactivity | 3.837 | 0.631 | 4.004 | 0.494 | 4.133 | 0.437 | 4.078 | 0.404 | 4.072 | .010* | .135 |
| Difficulties in emotional regulation | 2.184 | 0.518 | 1.962 | 0.498 | 1.990 | 0.424 | 1.758 | 0.413 | 15.788 | .000*** | .369 |
| Awareness | 2.518 | 0.673 | 2.218 | 0.522 | 2.100 | 0.498 | 2.036 | 0.590 | 12.318 | .000*** | .313 |
| Impulse | 1.882 | 0.637 | 1.696 | 0.524 | 1.629 | 0.501 | 1.564 | 0.375 | 4.175 | .008** | .134 |
| Non-acceptance | 2.104 | 0.607 | 1.736 | 0.703 | 1.809 | 0.593 | 1.518 | 0.536 | 9.126 | .000*** | .253 |
| Goals | 2.843 | 1.003 | 2.643 | 0.910 | 2.507 | 0.780 | 2.243 | 0.940 | 6.220 | .000*** | .187 |
| Clarity | 1.986 | 0.653 | 1.800 | 0.588 | 1.714 | 0.600 | 1.664 | 0.502 | 5.891 | .001** | .179 |
| Strategies | 1.771 | 0.594 | 1.679 | 0.571 | 1.629 | 0.532 | 1.521 | 0.448 | 2.474 | .067 | .084 |

*Note*. For these analyses, *n* = 29.

$\eta_p^2$ = .01 = low effect size; $\eta_p^2$ = .06 = medium effect size; $\eta_p^2$ = .14 = large effect size.

*p < .05.

**p < .01.

***p < .001.

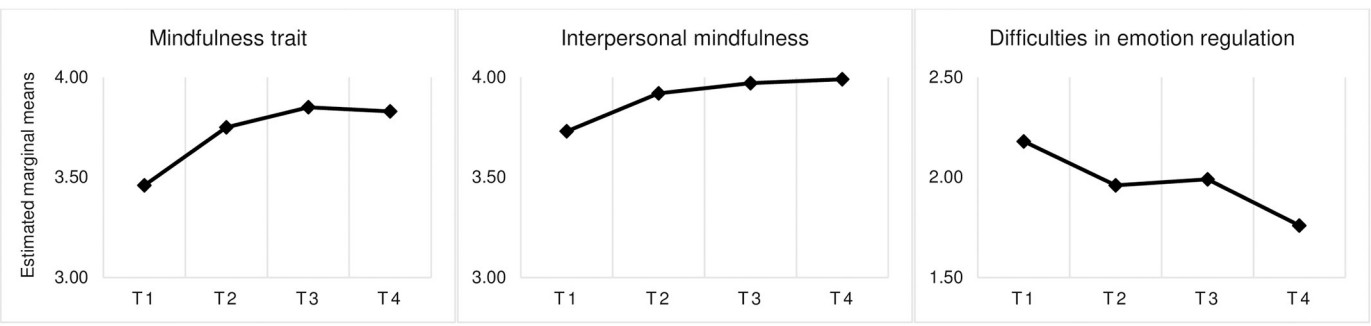

**Fig 2. Evolution of the estimated marginal means for the experimental group for each study variable.** Note. For these analyses, $n = 29$. T1 = pre-test; T2 = post-test; T3 = 3-month follow-up; T4 = 6-month follow-up.

T2 (post-test) ($t_{(3, 84)} = 4.191$; $p = .002$; 95% CI 3.267, 3.909), T1 and T3 (3-month follow-up) ($t_{(3, 84)} = 5.189$; $p < .001$; 95% CI 3.267, 4.007), and T1 and T4 (6-month follow-up) ($t_{(3, 84)} = 4.776$; $p < .001$; 95% CI 3.267, 3.986); in interpersonal mindfulness between T1 and T2 ($t_{(3, 84)} = 3.000$; $p = .037$; 95% CI 3.555, 4.089), T1 and T3 ($t_{(3, 84)} = 3.338$; $p = .015$; 95% CI 3.555, 4.124), and T1 and T4 ($t_{(3, 84)} = 3.394$; $p = 007$; 95% CI 3.555, 4.148), and in difficulties in emotion regulation between T1 and T2 ($t_{(3, 84)} = 3.524$; $p = .009$; 95% CI 1.769, 2.385), T1 and T3 ($t_{(3, 84)} = 4.645$; $p < .001$; 95% CI 1.732, 2.385), and T1 and T4 ($t_{(3, 84)} = 6.086$; $p < .001$; 95% CI 1.732, 2.385), as well as between T2 and T4 ($t_{(3, 84)} = 3.091$; $p = .029$; 95% CI 1.598, 2.155). The evolution of the estimated marginal means for each variable is illustrated in Fig 2.

## Qualitative results

Qualitative results indicated that the program had impacted participants in several ways, which were grouped into the following themes: (i) professional, (ii) personal, (iii) social, (iv) self-knowledge and (v) best practices. To ensure the anonymity of all participants, the names used are pseudonyms.

**Professional impact.** This theme groups the following subthemes: (i) value given to the program, (ii) self-regulation, (iii) interpersonal interactions and (iv) leadership. The value that the participants placed on the program shows the tremendous impact of the M×E, commenting that it was "essential" (Josep, technical director; P.5), "necessary" (Enric, coach; P.15), "more fundamental than the game itself" (Pol, technical director; P.8). The impact on self-regulation skills related to attention and emotions was remarkable. Highlighted element were observation and concentration:

> "Observing outside the technical contents, their body and facial expression, it was the most positive impact for me"
>
> (Pol, technical director; P.8).

> "The impact has been great both in practices and in games; I try to isolate myself from all the noise and all the other stimuli and be more focused so I can make better decisions"
>
> (Martí, coach; P.20).

On the other hand, M×E had an impact on emotional self-regulation allowing participants to recognize their emotional states and also to self-regulate their behavior: "it has had an impact empowering me to know when I am in a high stress, and to know how to stop" (Manel, project manager; P.9).

Regarding interpersonal interactions, some interviewees reported being better able to understand other colleagues' states and improve their communication skills:

"The program helped me to be present in every conversation

(Josep, technical director; P.5).

"I have improved a lot in these two months in the assertive communication"

(Manel, project manager; P.9).

Finally, participants referred to the importance and effect of carrying out the program to impact the people around them, including their teams:

"The fact that you can have a much healthier, more compassionate, accepting conversation with yourself, affects everyone around you"

(Toni, member of the Methodology Department; P.10)

"It is essential, because if you don't know how to manage yourself it is very difficult to manage others"

(David, technical director; P.2).

**Personal impact.**    The results were grouped into the following subthemes: (i) personal and professional life balance; (ii) improvements in personal relationships; and (iii) acquisition of healthy habits: "I have acquired the habit of meditating, read every night, exercise, which I had abandoned, eating better. . . it's like a whole, right? I am taking on habits that make me feel alive"(David, technical director; P.2).

**Social impact.**    Several participants highlighted group cohesion:

"The group cohesion that we have achieved. Seeing this more personal and emotional part, it's been good for all of us. It has helped me to know and work better with colleagues"

(Manel, project manager; P.9).

Finally, 'sense of belonging' was also one of the aspects emphasized by some participants.

**Self-knowledge.**    This theme was widely highlighted, with comments such as:

"The greatest impact the program has had is on self-knowledge"

(Joan, coach, P.3).

"It has given me a lot of knowledge, especially about myself"

(Sergi, technical director, P.4).

Self-knowledge was also transferred to a greater awareness of one's own leadership style:

"It has made me realize my leadership style"

(Andreu, technical director, P.6).

**Best practices.**   In addition to the themes presented, information regarding "best practices" on the program implementation emerged, emphasizing (i) its practical orientation, (ii) the informal practice, (iii) the professional practice and (iv) the 'safe space':

"Yes, on the field, when you said try to do this exercise, you mentioned the sound of the ball, and when I had this anchor it was easier be there and pay attention to what was happening"

(Carles, coach; P.7).

"For me, the most decisive thing is to create those environments where people expose themselves without fear; on the contrary, they expose their fears. . . This seems astounding to me, it's the game changer. This space and this bond"

(Toni, member of the Methodology Department; P.10).

## Discussion

The study's objective was to evaluate the impact of a new mindfulness-based program (M×E) on coaches, support staff, and technical directors of an elite football academy. All hypotheses were confirmed since participants in the M×E showed a significant increase in mindfulness trait and interpersonal mindfulness and a significant improvement in emotional regulation compared to the wait-list control group. Moreover, follow-up measurements of the experimental group showed increases up to 6 months in the three variables. The qualitative analysis reinforced the statistical results and revealed that the program positively impacted participants' professional and personal lives. Finally, "best practices" in the program administration were also extracted from the interviews.

The significant increase in mindfulness trait reinforced previous results in implementing mindfulness training for sports coaches [3]. The qualitative data showed how the participants transferred it to their jobs, highlighting moments of concentration and observation. These results are supported by Baltzell et al. (2015) [11], who suggested that mindfulness training could allow coaches to be more present and improve concentration on task-relevant cues. Moreover, the significant improvement in emotional regulation matches the results from Longshore and Sachs (2015) [3], where significantly greater emotional stability was shown after completing a mindfulness program. The ability to deal with emotions is considered a fundamental skill for coaches [33], with possible effects on well-being and performance [8].

Participants in the M×E showed a significant increase in levels of interpersonal mindfulness compared to the wait-list control group. According to Kee (2019) [16], coaches who are more aware, especially in terms of interpersonal attention, could gain deeper insight into their athletes, leading to better adjustments to task constraints. From the interviews, it also emerged that participants started to transfer what learned in during the mindfulness sessions to daily training, gaining deeper vision on players' behaviors: "Since the program, I began to be more aware and observe the behaviors of the players, what behaviors emerged from them. . . and came out with much more information" (Sergi, technical director). These results are similar to the study by Baltzell et al. (2015) [11], where coaches reported that they became more focused and able to understand the emotional and mental states of their players through participation in a mindfulness-based program. Another notable aspect was communication, one of the novelties of the M×E compared to previously implemented programs. This point is especially relevant since communication is considered one of the essential aspects of the coach's role, and

interventions that promote positive communication styles improve athletes' mental health [19], and maintain psychologically safe environments [20].

According to Côté and Gilbert (2009) [18], a coach's ability to maximize athletes' results is based on constant introspection and reviewing one's practice. Sports psychology has long recognized the importance of awareness, particularly self-awareness, for coaches to be more effective, as they can more effectively assess and respond to their players, can better see what effect their behaviors, feelings or communication have on their players, as well as understand what generates their thoughts, feelings and actions [21, 34]. M×E included the 'self-awareness' training plan to promote further this self-awareness and its transfer to the leadership style, which was widely highlighted by participants: "The greatest impact the program has had is on self-knowledge" (Joan, coach).

Importantly, mindfulness is considered a way to improve performance and overall well-being [24]. The balance between work and personal life found in this study is considered a key protective factor for having good mental health [35], and these results reinforce those of Longshore and Sachs (2015) [3]. Furthermore, the acquisition of healthier habits was also noticeable during the interviews. Importantly, researchers agree that coaches often lack self-care, leading to high levels of stress and burnout [36]. Finally, M×E's participants reported how the program had impacted their relationships. This finding reinforces the previous results of Baltzell et al. (2015) [11] and Longshore and Sachs (2015) [3], where coaches reported improved social relationships. Similarly, participants also emphasized group cohesion and connection with other participants, as previously highlighted by Spencer et al. (2019) [37]. The 'sense of belonging' also emerged from the interviews and is relevant since belonging and having close social relationships are fundamental human needs related to well-being [38]. The results of this study add to other research that shows that mindfulness contributes to a better quality of life [39] and reinforce that incorporating mindfulness for coaches can benefit them beyond professional development but their overall well-being.

To end with, one of the main objectives of this study was to detect best practices in implementing such programs. For instance, the practical orientation of the M×E was one of the most valued elements. Informal practice and STOP moments regarding the practice and training plan were frequently mentioned in the interviews. The professional practice of the training plan designed specifically for this program also had a great impact and constitutes one of the best practices of the M×E as well as the self-awareness practice. The in-class group meditation was appreciated as a way of connection between participants, something also found in Baltzell et al. (2015) [11], where the coaching staff stated that meditating with the athletes deepened the level of closeness that the coaches felt with their players.

The 'safe space' effects were, without a doubt, one of the most relevant impacts of the M×E. According to Edmondson (1999) [40], creating an environment that is perceived as psychologically safe reflects a climate of interpersonal trust, mutual respect, and acceptance, which facilitates active participation. Various research in work contexts has reinforced the positive outcomes of psychological safety [41], with positive associations with team performance [42] and well-being [43].

Finally, considering the voluntary nature of the program, some preconceived ideas of the participants about the value of mindfulness could have influenced their evaluation. Also, most participants knew the leader and expressed their satisfaction which could have influenced their evaluations of the overall program, which is consistent with previous research (Baltzell et al., 2015) [11]. However, Minkler et al. (2020) [44] found benefits when someone led the program with ongoing contact with the team. Considering the positive results of this study, the fact that the program leader is part of the usual dynamic is a good practice.

The present study has some limitations that should be considered. The quasi-experimental design makes it impossible to infer that the program was solely responsible for the observed improvements. Another limitation was due to the fact that the Interpersonal Mindfulness Scale (IMS) did not receive previous psychometric validation. However, best practices for questionnaire translations were followed for adapting the scale for use within Spanish speakers [45]. Another limitation was the use of an unvalidated translation of the. Finally, the leading researcher was the one who carried out the entire process: providing training, conducting the interviews, managing the data from the questionnaires and performing the statistical analyses. Future research could prevent potential bias by training a trainer who is not part of the research team.

The present research suggests future research directions. For instance, the relevance of the follow-up and qualitative data should be necessary in these types of studies. The success of the online format motivates its continued improvement and research due to its clear time-space advantages. However, future research should explore the M×E's impact of the in-person format. Considering the top-down perspective from which the M×E proposal is born, it would be important to include evaluations of the program's impact on athletes indirectly. Mindfulness-based interventions should aim to be longer and more integrated into the team and club dynamics. Finally, given that the results of the present study indicate the suitability and effectiveness of applying the M×E, it is justified that future research replicates the study design in other groups and contexts. The M×E must continue to adapt in its application to practice, identifying more scenarios in which coaches, technical directors and other support staff can use the principles and strategies of mindfulness to improve not only their performance and well-being but also athletes' environment.

## Conclusions

This is the first study to evaluate the M×E impact on coaches, support staff and Technical Directors' mindfulness ability, interpersonal mindfulness, and emotion regulation. Participants in the program significantly improved their mindfulness trait and interpersonal mindfulness and significantly decreased their difficulties in emotional regulation compared to the wait-list control group. Qualitative results showed the transference in emotional and attentional self-management skills associated with performance, interpersonal skills associated with leadership and healthy habits and professional and personal life balance associated with well-being. Follow-up measurements indicate that a reinforcement action should be done at least after six months. Furthermore, the results made it possible to reinforce and detect best practices in implementing a mindfulness-based program for this population. The positive results and the unanimous satisfaction with the M×E program postulate it as a reliable candidate for training coaches, support staff and sports leaders and invite sports organizations to consider its implementation.

## Supporting information

**S1 File.**
(DOCX)

**S1 Data.**
(XLSX)

## Acknowledgments

We extend my heartfelt appreciation to everyone who contributed to the success of this scientific project.

## Author Contributions

**Conceptualization:** Aleix Gibert.

**Data curation:** Aleix Gibert.

**Formal analysis:** Aleix Gibert, Antoni Planas.

**Investigation:** Aleix Gibert.

**Methodology:** Aleix Gibert.

**Project administration:** Aleix Gibert.

**Resources:** Aleix Gibert.

**Software:** Aleix Gibert.

**Supervision:** Carlota Torrents.

**Writing – original draft:** Aleix Gibert.

**Writing – review & editing:** Carlota Torrents.

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
