## [Decision Letter · Decision Letter 0]

30 Apr 2024

PONE-D-24-07175Impact of mindfulness on football coaches: a mixed designPLOS ONE

Dear Dr. Gibert,

Thank you for submitting your manuscript to PLOS ONE. After careful consideration, we feel that it has merit but does not fully meet PLOS ONE’s publication criteria as it currently stands. Therefore, we invite you to submit a revised version of the manuscript that addresses the points raised during the review process.

We look forward to receiving your revised manuscript.

Kind regards,

Ender Senel, PhD

Academic Editor

PLOS ONE

Journal Requirements:

Reviewers' comments:

Reviewer's Responses to Questions

**Comments to the Author**

1. Is the manuscript technically sound, and do the data support the conclusions?

Reviewer #1: Yes

2. Has the statistical analysis been performed appropriately and rigorously? 

Reviewer #1: Yes

3. Have the authors made all data underlying the findings in their manuscript fully available?

Reviewer #1: Yes

4. Is the manuscript presented in an intelligible fashion and written in standard English?

Reviewer #1: Yes

5. Review Comments to the Author

Reviewer #1: The manuscript (MS) is generally well-written, and appropriate methods have been implemented in the design, ethics, intervention, and analyses.

I would like the authors to address:

(1) Why the wait list control group received the program just after the pre- and post- evaluation? This does not allow to observe possible differences in the two follow-ups…

(2) In the whole MS, the authors use the expression “decrease in difficulties in emotion regulation”. This is, in my opinion, counterintuitive and I would generally refer to “improvement in emotion regulation”.

Please, see my specific comments below:

Abstract

- “(MxE)” please specify what MxE is

- “A significant decrease in difficulties in emotion regulation” I would write “significant improvement in emotion regulation”

Introduction (No particular comments)

Method

Why the wait list control group received the program just after the pre- and post- evaluation? This does not allow to observe possible differences in the two follow-ups… Please, provide a reason for this

Lines 109-114, the authors state that 48.3% of participants meditated sometimes; and that the sample consisted of coaches (39.7%), technical directors (12.1%) and coordinator (34.5%). Were these proportions maintained in the experimental and in the control group?

Line 132, at this point I presume MxE stands for Mindfulness for Enhancement? Please, specify at the beginning of the manuscript and in the abstract

Line 147. The table is very well-done and clarifies the whole MxE programme.

Lines 192-193. “translated the questionnaire”? Then the authors can mention the fact that the questionnaire was not previously validated in Spanish language in the Limitation section at Lines 441-447.

Lines 195. “of THE Difficulties IN Emotion Regulation Scale”?

Lines 199-200. Check verb tenses, please.

Lines 201-202. Please, expand this part with the description of the practice-log.

Line 221. Standard Deviation (SD)?

Results:

Line 280. Table 3 should mention somewhere that these data were relating to the sole experimental group.

Lines 345-348. If this theme “was widely highlighted”, the authors could expand a bit more this paragraph.

Discussion:

Lines365-366. I understand that the questionnaire was measuring “difficulties in emotion regulation” and that there was a decrease in these dimensions. However, it seems odd to read “a significant decrease in difficulties in emotional regulation”. I would invite the authors to consider writing “a significant improvement in emotion regulation” here and in the abstract. All fine in all the other parts I have read so far.

Line 376. Again “significant decrease in difficulties in emotional regulation”. Please, consider rewording this sentence.

Lines 383-385: “The transfer of mindfulness …” this sentence is a bit unclear to me, I would reword it as follows: “From the interviews, it also emerged that participants started to transfer what learned in during the mindfulness sessions to daily training, gaining deeper vision on players’ behaviours: “…”

Line 443-444. “A limitation was due to the fact that the scale did not receive previous psychometric validation. However, best practices for questionnaire translations were followed for adapting the scale for use within Spanish speakers.” Or something along these lines. Please, add a reference to support that the adopted procedure was a best practice for questionnaire translations.

Line 464. “on coaches, support staff and technical directors’ mindfulness ability and emotional stability”?.

6. PLOS authors have the option to publish the peer review history of their article (what does this mean?). If published, this will include your full peer review and any attached files.

Reviewer #1: No

---

## [Author Response · Author response to Decision Letter 0]

3 Jun 2024

Response to Reviewers

Reviewer #1: The manuscript (MS) is generally well-written, and appropriate methods have been implemented in the design, ethics, intervention, and analyses.

I would like the authors to address:

(1) Why the wait list control group received the program just after the pre- and post- evaluation? This does not allow to observe possible differences in the two follow-ups…

We are aware of the methodological limitations of this and it was forced by the Club who wanted to provide the program to as many staff as possible and as soon as possible. In order to get the whole study done, we had to agree on this situation.

(2) In the whole MS, the authors use the expression “decrease in difficulties in emotion regulation”. This is, in my opinion, counterintuitive and I would generally refer to “improvement in emotion regulation”.

Agreed and changed in the manuscript.

Please, see my specific comments below:

Abstract

- “(MxE)” please specify what MxE is

 Done.

- “A significant decrease in difficulties in emotion regulation” I would write “significant improvement in emotion regulation”

Done.

Introduction (No particular comments)

Method

Why the wait list control group received the program just after the pre- and post- evaluation? This does not allow to observe possible differences in the two follow-ups… Please, provide a reason for this.

We are aware of the methodological limitations of this and it was forced by the Club who wanted to provide the program to as many staff as possible and as soon as possible. In order to get the whole study done, we had to agree on this situation.

Lines 109-114, the authors state that 48.3% of participants meditated sometimes; and that the sample consisted of coaches (39.7%), technical directors (12.1%) and coordinator (34.5%). Were these proportions maintained in the experimental and in the control group?

Yes, proportions were very similar in both groups.

Line 132, at this point I presume MxE stands for Mindfulness for Enhancement? Please, specify at the beginning of the manuscript and in the abstract

Done.

Line 147. The table is very well-done and clarifies the whole MxE programme.

Lines 192-193. “translated the questionnaire”? Then the authors can mention the fact that the questionnaire was not previously validated in Spanish language in the Limitation section at Lines 441-447.

Corrected. 

Lines 195. “of THE Difficulties IN Emotion Regulation Scale”?

Corrected. 

Lines 199-200. Check verb tenses, please.

Corrected. 

Lines 201-202. Please, expand this part with the description of the practice-log.

Expanded.

Line 221. Standard Deviation (SD)?

Corrected. 

Results:

Line 280. Table 3 should mention somewhere that these data were relating to the sole experimental group.

Corrected. 

Lines 345-348. If this theme “was widely highlighted”, the authors could expand a bit more this paragraph.

Expanded.

Discussion:

Lines 365-366. I understand that the questionnaire was measuring “difficulties in emotion regulation” and that there was a decrease in these dimensions. However, it seems odd to read “a significant decrease in difficulties in emotional regulation”. I would invite the authors to consider writing “a significant improvement in emotion regulation” here and in the abstract. All fine in all the other parts I have read so far.

Corrected. 

Line 376. Again “significant decrease in difficulties in emotional regulation”. Please, consider rewording this sentence.

Corrected. 

Lines 383-385: “The transfer of mindfulness …” this sentence is a bit unclear to me, I would reword it as follows: “From the interviews, it also emerged that participants started to transfer what learned in during the mindfulness sessions to daily training, gaining deeper vision on players’ behaviours: “…”

Corrected. 

Line 443-444. “A limitation was due to the fact that the scale did not receive previous psychometric validation. However, best practices for questionnaire translations were followed for adapting the scale for use within Spanish speakers.” Or something along these lines. Please, add a reference to support that the adopted procedure was a best practice for questionnaire translations.

Corrected. 

Line 464. “on coaches, support staff and technical directors’ mindfulness ability and emotional stability”?.

Corrected.

---

## [Decision Letter · Decision Letter 1]

25 Jun 2024

Impact of mindfulness on football coaches: a mixed design

PONE-D-24-07175R1

Dear Dr. Gibert,

We’re pleased to inform you that your manuscript has been judged scientifically suitable for publication and will be formally accepted for publication once it meets all outstanding technical requirements.

Kind regards,

Ender Senel, PhD

Academic Editor

PLOS ONE

Additional Editor Comments (optional):

Reviewers' comments:

Reviewer's Responses to Questions

**Comments to the Author**

1. If the authors have adequately addressed your comments raised in a previous round of review and you feel that this manuscript is now acceptable for publication, you may indicate that here to bypass the “Comments to the Author” section, enter your conflict of interest statement in the “Confidential to Editor” section, and submit your "Accept" recommendation.

Reviewer #1: All comments have been addressed

2. Is the manuscript technically sound, and do the data support the conclusions?

Reviewer #1: Yes

3. Has the statistical analysis been performed appropriately and rigorously? 

Reviewer #1: Yes

4. Have the authors made all data underlying the findings in their manuscript fully available?

Reviewer #1: (No Response)

5. Is the manuscript presented in an intelligible fashion and written in standard English?

Reviewer #1: (No Response)

6. Review Comments to the Author

Reviewer #1: Interpersonal repeated twice among the keywords

All my comments have been addressed and the Manuscript is well-written

7. PLOS authors have the option to publish the peer review history of their article (what does this mean?). If published, this will include your full peer review and any attached files.

Reviewer #1: No
